# Reduced Vessel Density in the Mid-Periphery and Peripapillary Area of the Superficial Capillary Plexus in Non-Proliferative Diabetic Retinopathy

**DOI:** 10.3390/jcm11030532

**Published:** 2022-01-21

**Authors:** Amira Chaher, Franck Fajnkuchen, Sandrine Tabary, Audrey Giocanti-Aurégan

**Affiliations:** 1Ophthalmology Department, Avicenne Hospital, 125 rue de Stalingrad, 93000 Bobigny, France; amirachaher@outlook.fr (A.C.); franck.fajnkuchen@gmail.com (F.F.); 2Centre d’Imagerie et de Laser, 11 rue Antoine Bourdelle, 75015 Paris, France; sandrine.tabary@cil-paris.fr

**Keywords:** diabetic retinopathy, optical coherence tomography angiography, retinal non-perfusion, vessel density, skeleton density

## Abstract

Our aim in this study was to assess the vessel density (VD) and vessel skeleton density (VSD) in the nasal area of the superficial capillary plexus (SCP) of diabetic subjects without diabetic retinopathy (DR), or in those with a non-proliferative diabetic retinopathy (NPDR), and to evaluate the relationship between the VD and VSD and the severity of DR. In this prospective study, the VD and VSD in the SCP were measured and analyzed on 6 × 6-mm macular and nasal optical coherence tomography angiography scans. The three concentric circles of the Early Treatment of Diabetic Retinopathy Study (ETDRS) grid were used and divided into zones numbered from 1 to 9 in the macular area and from 1 to 8 in the nasal area. The VD was significantly lower in the nasal peripapillary area (*p* = 0.0028), and both the VD and VSD were significantly lower in the macular area (*p* = 0.0131 and *p* = 0.0132, respectively) in patients with more severe DR. The SD was significantly lower in zones 5 (*p* = 0.0315) and 6 (*p* = 0.0324) in the nasal grid in patients with more severe DR. We showed a lower superficial capillary flow in the nasal periphery and peripapillary area in patients with more severe DR.

## 1. Introduction

Optical coherence tomography angiography (OCT-A) is a depth-resolved imaging modality, allowing the visualization of vessels by detecting the decorrelation signal between successive OCT B-scans due to blood flow motion. Unlike fluorescein angiography, it is a non-invasive, dyeless examination. OCT-A allows for the assessment of the retinal vasculature, and the software can separate the superficial vascular plexus (SVP) from the deep vascular complex (DVC), which is useful for studying diabetic retinopathy (DR).

Several studies have reported a reduced macular capillary density in the superficial capillary plexus (SCP) and/or in the deep capillary plexus (DCP) on OCT-A in patients with DR compared to healthy controls, and most studies have found a linear correlation with disease severity when the healthy, without DR, non-proliferative DR (NPDR) and proliferative DR (PDR) groups were compared [1,2,3,4,5]. Moreover, the foveal avascular zone is larger in diabetic eyes compared to control eyes [6,7,8,9]. However, the automatic segmentation of OCT-A may be limited by the presence of a macular edema [1]. Numerous artifacts may alter macular perfusion analysis and prevent visualization of the actual anatomy. To avoid these limitations, we analyzed two parameters that are easily and automatically available on most OCTA devices, the vessel density (VD) and vessel skeleton density (VSD) in the nasal area of the SCP, in a cohort of diabetic subjects, knowing that this area is usually the most ischemic part of the retina (according to the Early Treatment of Diabetic Retinopathy Study (ETDRS)). The aim of this study was to assess the relationship between nasal retinal non-perfusion and the severity of DR.

## 2. Materials and Methods

This prospective study was conducted in accordance with the tenets of the Declaration of Helsinki and was approved by the Ethics Committee of the French Society of Ophthalmology (IRB 00,008,855 Société Française d’Ophtalmologie IRB#1).

Study population: The records of consecutive diabetic patients who consulted our ophthalmology practice (Centre Ophtalmologique d’Imagerie et de Laser, CIL, Paris, France) between December 2016 and January 2021 were reviewed.

All subjects underwent a fundus examination and ultra-wide-field retinal imaging using the Optos scanning laser ophthalmoscope imaging system (Optos PLC, Dunfermline, Scotland, UK) to assess the stage of DR. Images were analyzed by two experienced retina specialists (AC, FF) using the simplified American Academy of Ophthalmology DR grading scale [10].

The characteristics of diabetes (type and HbA1c level) and patients’ comorbidities were collected. In addition to OCT-A measurements, the central macular thickness (CMT) was recorded. Our groups of analysis were adjusted for smoking status [11].

We included diabetic patients without DR, or with mild, moderate or severe NPDR. Additional prerequisites were OCT-A images without significant movement or shadow artifacts, and a signal strength index (SSI) greater than 60.

Exclusion criteria were eyes with PDR or with other retinal vascular diseases such as retinal vein occlusion, glaucoma and ocular hypertension, and eyes previously treated with an anti-VEGF or corticosteroid intravitreal injection and macular or panretinal photocoagulation. Eyes with history of vitreous surgery, and those with ocular ischemic syndrome were excluded. Patients with general co-morbidities such as cardiovascular diseases or coagulation disorders were also excluded.

### 2.1. OCT Angiography (OCT-A) System Setup and Analysis

OCT-A examinations were performed with the CIRRUS HD-OCT 5000 with AngioPlex software (Carl Zeiss Meditec, Oberkochen, Germany, software version 11.0.0.29946). Flow images from repeated B-scans were obtained using the OCT microangiography complex (OMAGC) algorithm. Measurements of the vessel density (VD) and vessel skeleton density (VSD) in the SCP were analyzed on the 6 × 6-mm area. The reference plane for the superficial plexus was defined as the area from the inner limiting membrane to the inner plexiform layer.

During OCT data acquisition, a fixation point in the center of the view was first used as the target to be located by the subject for the macular acquisition. Then, the fixation point was moved in the nasal area to obtain the nasal acquisition.

The 3 concentric circles of the ETDRS chart can be divided into 9 zones, as shown in Figure 1. The inner circle was centered on the fovea for the macular acquisition. The outermost zone was centered on the optic disc for the nasal acquisition (Figure 2).

### 2.2. Statistical Analysis

Quantitative variables are presented as a mean and standard deviation (SD). A Kruskal–Wallis one-way analysis of variance (ANOVA) was used to compare the nasal and macular VD and SD between the DR groups. A *p*-value < 0.05 was considered statistically significant. All statistical analyses were performed using GraphPad Prism version 9.2.0 (GraphPad Software, San Diego, CA, USA).

## 3. Results

### 3.1. Study Population

A total of 97 eyes of 56 diabetic subjects were included in the study. The subjects’ mean age was 62 years (range: 12–90 years). There were 44 men (78.6%) and 12 women (21.4%). Fifteen eyes were excluded from the analysis: three due to poor OCT-A image quality, one due to the presence of Irvine–Gass syndrome, one due to a high intraocular pressure with retinal nerve fiber layer defect, two due to the presence of central vein occlusion, two due to a previous intravitreal injection for diabetic macular edema, one due to previous panretinal photocoagulation, three due to the absence of acquisition and two due to the presence of PDR (Figure 3). Pertinent demographics are shown in Table 1.

Because of the small number of subjects with mild DR, the eyes with mild DR and with moderate DR were grouped together. The mean best corrected visual acuity was 77 letters (range: 75–85 letters) in the no DR group, 78.4 (range 72–85) in the mild and moderate non-proliferative DR (NPDR) group, and 78 letters (range 73–85 letters) in the severe NPDR group (*p* = 0.95).

### 3.2. Quantitative Analysis

#### 3.2.1. Nasal Area

Regarding VD, only area 4, immediately adjacent to the optic nerve head, showed significantly lower values with DR severity (Figure 4) (*p* = 0.0028). The other areas tended to show a lower perfusion in eyes with more severe DR but without reaching significance (Table 2).

The analysis was repeated including only one eye per patient (the right eye when DR was equally severe in both eyes, and the eye with the worst stage of DR otherwise).

On 56 eyes of 56 patients, we confirmed that the VD showed significantly lower values with DR severity in zone 4 (*p* = 0.04) (Appendix A).

The VSD was significantly lower in eyes with more severe DR in zones 5 (Figure 5) (*p* = 0.0315) and 6 (Figure 6) (*p* = 0.0324). The data in zones 7 and 8 tended to show a lower perfusion in eyes with more severe DR, but without reaching significance (*p* = 0.1567 and *p* = 0.1860, respectively).

In the subgroup including only one eye per patient, the VSD was also significantly lower in eyes with more severe DR in zones 5 (*p* = 0.004) (Appendix A) and 6 (*p* = 0.01) (Appendix A).

#### 3.2.2. Macular Area

Only zone 8 of the macular grid, immediately adjacent to the optic nerve head, showed lower VD and SD in eyes with more severe DR, (*p* = 0.0132 for the SD, and *p* = 0.0131 for the VD).

## 4. Discussion

We showed a relationship between the VD and VSD in the nasal area of the SCP and the severity of DR.

The originality of this study lies in two points. First, we assessed the retinal perfusion in the nasal area rather than in the macular area using OCT-A. Second, to our knowledge, this is the only study to compare perfusion data between eyes with the different stages of NPDR, except for eyes with mild NPDR that were grouped with eyes with moderate NPDR. Most studies have compared diabetic subjects to healthy subjects, or a group of subjects with NPDR of all stages to a group of subjects with PDR.

We chose to study the nasal area because the ETDRS has already shown that capillary loss may be more marked in the nasal part of the retina and of the optic nerve head, which corresponds exactly to the area we studied [12,13]. We could have studied the flow data in another area in the periphery, but the nasal area is more convenient. An OCT-A acquisition in the inferior or superior part of the retina, for example, would have been more challenging. The method has been shown to be reproducible using the motion tracking of the Cirrus 5000 from Zeiss [14].

Previous studies have already reported a heterogeneous perfusion on fluorescein angiography and wide-field OCT-A (Figure 7) in diabetic subjects with a predominance of capillary non-perfusion in the mid-peripheral retina [15,16,17]. In 1981, Shimizu et al. used a 130° fluorescein angiography montage technique to show that the extent of capillary non-perfusion in the mid-periphery increased with DR severity. More recently, Alibhai et al. showed a correlation between the mean percentage of non-perfused area and the severity of DR on OCT-A using wide-field 12 × 12-mm images. The mean percentage of non-perfused area was lower in eyes without DR; it increased in eyes with NPDR, and more strongly in eyes with PDR [16].

Several studies have suggested that the capillary perfusion could be preserved around the macula at the expense of the periphery for a period of time in eyes with DR [18,19,20]. This preservation of the macular area could be due to the secretion of biochemical factors by the ischemic retina that would particularly protect the central retina. The other hypothesis is that anatomical differences in retinal microcirculation between the different regions of the retina could explain the development of compensatory mechanisms in macular microcirculation.

Among the three most peripheral locations of the nasal grid (zones 6, 7 and 8), we showed a significantly lower perfusion in eyes with more severe DR only in zone 6, while only a trend was observed in zones 7 and 8, as shown in Figure 6.

Such a decrease in capillary density in the periphery has already been described in healthy subjects. Carlo Lavia et al. have shown a decreased density in the periphery of the SCP at about 8 mm from the fovea in healthy subjects. This location corresponds to the area where the intermediate capillary plexus vanishes in the periphery [21]. This threshold of 8 mm (about 8 mm temporally and 10 mm nasally, if the optic nerve head is not taken into account) corresponds to zones 6, 7 and 8 of the grid used in our study (Figure 8), that is, the outermost zones of the ETDRS grid moved nasally (Figure 2 and Figure 8). Thus, this peripheral zone could be more vulnerable to ischemia.

This heterogeneity of retinal perfusion seems to be comparable to that of the neuronal tissue. The density in the different capillary plexuses indeed correlates with the neuronal volume, the thickness of the cell layers and the metabolic activity of the tissue, which decreases with the distance from the fovea [22]. Interestingly, in our study, there was a significant relationship between vascular density and the severity of DR in the peripapillary areas (zone 4 of the nasal grid and zone 8 of the macular grid). These areas correspond to the areas where the radial peripapillary capillary plexus (RPCP) is present. The RPCP is not individualized by Cirrus segmentation, but it is localized in the optic fiber layer that is included in the segmentation of the SCP. The reduction in RPCP during DR progression has been assessed in previous studies [23,24].

The early impairment of the RPCP could be explained by the different anatomy of this capillary network. On the one hand, the greater path length of these capillaries involves a greater resistance than in the rest of the retinal capillary network. In addition, the RPCP derives from small, single peripapillary arterioles and has a few intercommunications with the adjacent capillaries, which reduces the possibilities of collateral supply [25].

This study has some limitations, including the absence of assessment of the DCP. Previous studies have not always shown a correlation between the DCP and the severity of DR [26]. Nevertheless, the SCP seems to be the most likely to correlate with ischemia [27] and the easiest plexus to be studied, because it is devoid of projection artifacts. Another limitation is the lack of a control group including healthy subjects. It could be assumed that a greater capillary perfusion would be observed in healthy subjects, as shown in the macular area, but it would have been interesting to make the comparison in the nasal periphery.

## 5. Conclusions

This study highlighted the value of OCT-A as a screening and monitoring tool for DR. The lower perfusion found in the nasal mid-periphery and peripapillary area showed the heterogeneity of the retinal microcirculation.

It would be interesting to investigate these data in healthy subjects and in subjects with PDR, as well as in other capillary plexuses.

## Figures and Tables

**Figure 1 jcm-11-00532-f001:**
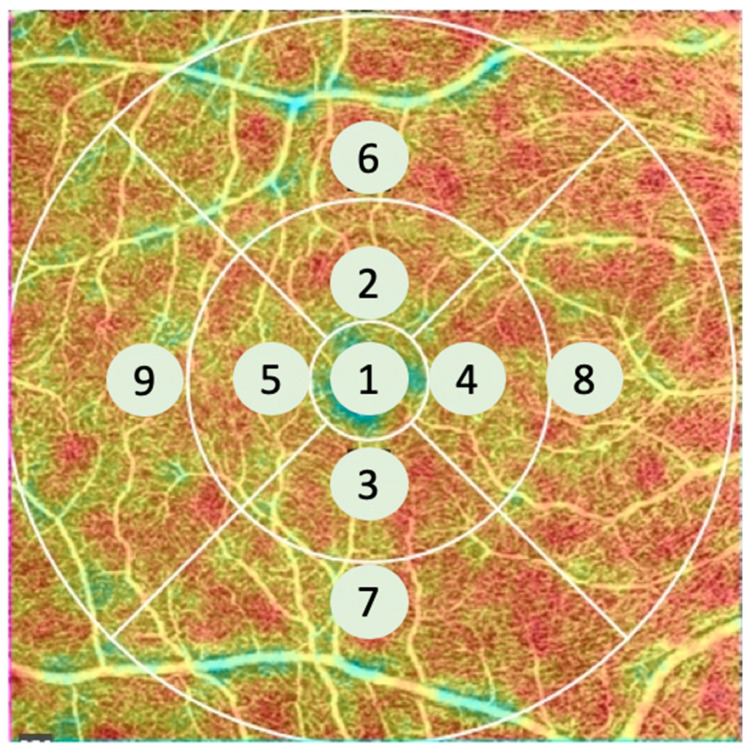
Example of a right eye. The 9 areas of the ETDRS (Early Treatment of Diabetic Retinopathy Study) grid were numbered so that area 8 was the most adjacent and area 9 was the most temporal to the optic nerve head.

**Figure 2 jcm-11-00532-f002:**
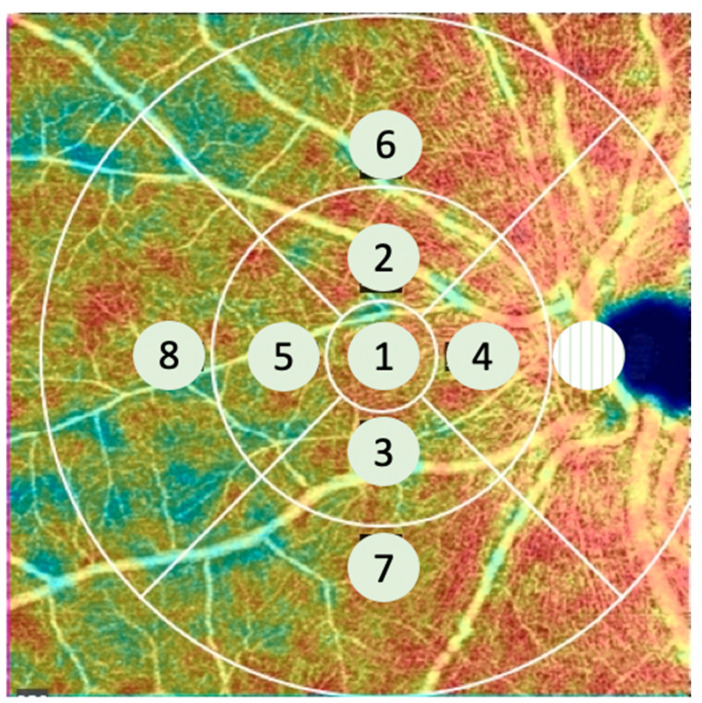
Example of a left eye. The ETDRS (Early Treatment of Diabetic Retinopathy Study) grid was placed nasally using the optic nerve head as a marker. Zone 4 was the most adjacent to the optic nerve head.

**Figure 3 jcm-11-00532-f003:**
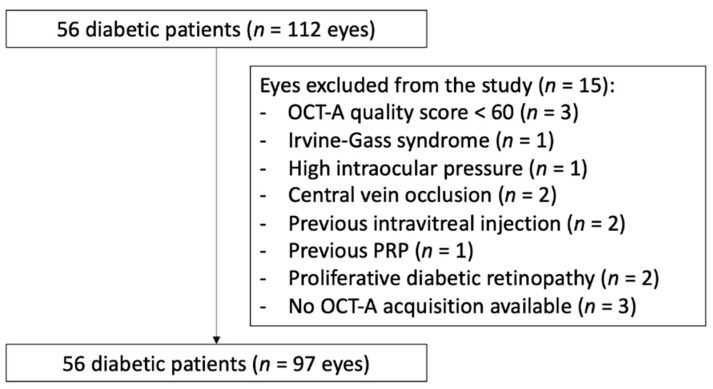
Flow chart of the study.

**Figure 4 jcm-11-00532-f004:**
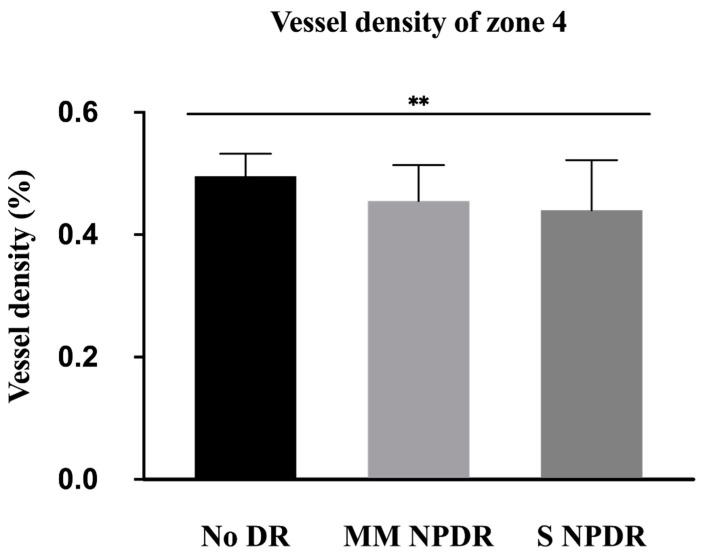
Mean vessel density with standard deviation for each group (without DR, mild and moderate NPDR, and severe NPDR) in zone 4. ** *p* < 0.01. DR, diabetic retinopathy; MM NPDR, mild and moderate non-proliferative diabetic retinopathy; S NPDR, severe non-proliferative diabetic retinopathy.

**Figure 5 jcm-11-00532-f005:**
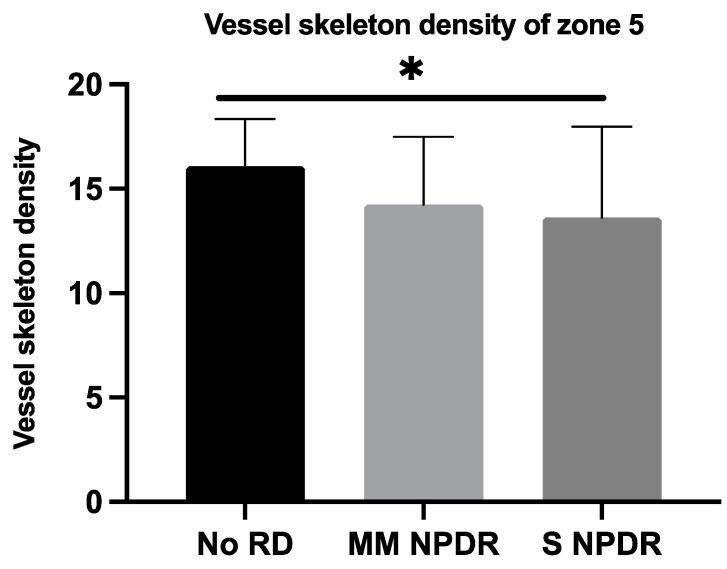
Mean vessel skeleton density with standard deviation for each group (without DR, mild and moderate NPDR, and severe NPDR) in zone 5. * *p* < 0.05. DR, diabetic retinopathy; MM NPDR, mild and moderate non-proliferative diabetic retinopathy; S NPDR, severe non-proliferative diabetic retinopathy.

**Figure 6 jcm-11-00532-f006:**
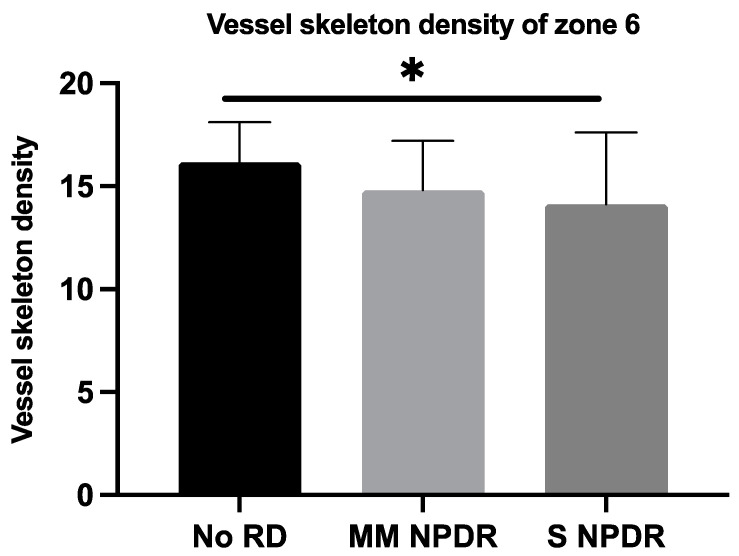
Mean vessel skeleton density with standard deviation for each group (without DR, mild and moderate NPDR, and severe NPDR) in zone 6. * *p* < 0.05. DR, diabetic retinopathy; MM NPDR, mild and moderate non-proliferative diabetic retinopathy; S NPDR, severe non-proliferative diabetic retinopathy.

**Figure 7 jcm-11-00532-f007:**
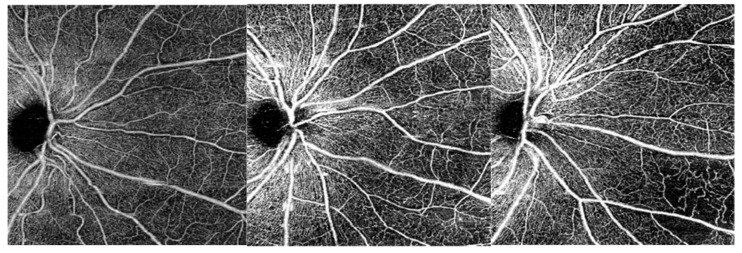
OCT-A C-scans of the superficial capillary plexus (SCP). From left to right: examples of a patient without DR, a patient with mild NPDR and a patient with severe NPDR. The vessel density in the SCP progressively decreases with DR severity, especially in the most peripheral area. DR, diabetic retinopathy; NPDR, non-proliferative diabetic retinopathy.

**Figure 8 jcm-11-00532-f008:**
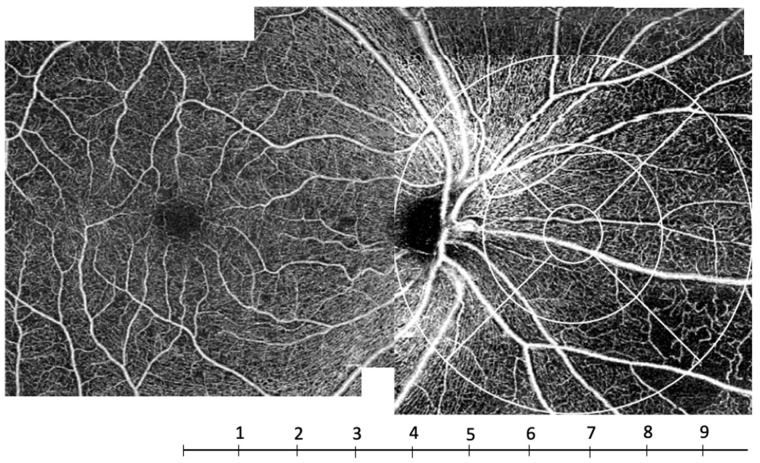
OCTA C-scan montage of the superficial capillary plexus (SCP) in a subject with severe non-proliferative diabetic retinopathy. The scale under the image indicates the distance to the fovea in millimeters. The capillary rarefaction predominated at a distance of about 8 mm from the fovea corresponding to zones 6, 7 and 8.

**Table 1 jcm-11-00532-t001:** Demographics of our cohort.

	No DR	Mild NPDR	Moderate NPDR	Severe NPDR
Patients/Eyes, *n*	16/29	6/10	22/38	12/20
Male/Female, *n*	13/3	4/2	15/7	10/2
Age, years				
Mean ± SD	61.6 ± 19.1	70.7 ± 12.4	62.2 ± 13.7	59.3 ± 13.4
Range	12–85	52–82	35–90	41–80
HbA1c, %				
Mean ± SD	7.5 ± (1.4)	6.8 ± (0.5)	7.1 (0.8)	7.2 (1.0)
Range	6.1–12.0	6.3–7.5	6–8.5	6–9
CMT, µm ± SD	276.0 ± 33.8	264.0 ± 20.8	300.1 ± 93.9	326 ± 76
Central edema, no. eyes (%)	3 (10.3)	0 (0)	6 (15.8)	13 (66.7)
SS nasal area	8.6 (1.1)	8.2 (1.1)	8.4 (0.9)	8.0 (1.0)
SS macular area	8.6 (1.1)	8.3 (1.3)	8.9 (1.1)	8.0 (2.0)

DR, diabetic retinopathy; NPDR, non-proliferative diabetic retinopathy; HbA1c, glycated hemoglobin; CMT, central macular thickness; SS, signal strength.

**Table 2 jcm-11-00532-t002:** Vessel density (VD) and vessel skeleton density (VSD) in each area of the ETDRS grid in the nasal area. DR, diabetic retinopathy; MM NPDR, mild and moderate non-proliferative diabetic retinopathy; S NPDR, severe non-proliferative diabetic retinopathy.* *p* < 0.05.

	No DR	MM NPDR	S NPDR	*p **
VDN1	0.42 ± 0.07	0.39 ± 0.08	0.39 ± 0.11	0.1695
VDN2	0.43 ± 0.07	0.42 ± 0.06	0.39 ± 0.12	0.7567
VDN3	0.39 ± 0.08	0.37 ± 0.09	0.36 ± 0.10	0.1563
VDN4	0.47 ± 0.08	0.46 ± 0.06	0.44 ± 0.08	0.0028 *
VDN5	0.39 ± 0.07	0.35 ± 0.09	0.35 ± 0.11	0.7059
VDN6	0.40 ± 0.07	0.38 ± 0.07	0.37 ± 0.09	0.0609
VDN7	0.36 ± 0.08	0.33 ± 0.09	0.34 ± 0.10	0.4248
VDN8	0.34 ± 0.08	0.32 ± 0.08	0.32 ± 0.10	0.8596
VSDN1	16.96 ± 2.62	16.35 ± 2.53	15.51 ± 4.15	0.3682
VSDN2	16.79 ± 2.39	16.37 ± 1.99	14.87 ± 4.29	0.3748
VSDN3	15.69 ± 2.66	14.77 ± 3.50	14.33 ± 3.89	0.3012
VSDN4	18.08 ± 2.85	18.18 ± 1.86	16.99 ± 2.95	0.1023
VSDN5	15.84 ± 2.58	14.16 ± 3.21	13.63 ± 4.35	0.0315 *
VSDN6	15.83 ± 2.60	15.14 ± 2.34	14.13 ± 3.48	0.0324 *
VSDN7	14.26 ± 3.01	13.13 ± 3.23	13.02 ± 3.59	0.1567
VSDN8	13.86 ± 3.07	12.90 ± 3.03	12.65 ± 4.08	0.1860

## Data Availability

Data are available upon request at Audrey.giocanti@aphp.fr.

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
