# Peer review of "Reduced Vessel Density in the Mid-Periphery and Peripapillary Area of the Superficial Capillary Plexus in Non-Proliferative Diabetic Retinopathy"

_jcm, 2022, doi:10.3390/jcm11030532_

Round 1

Reviewer 1 Report

The phrase vessel skeleton density (VSD) instead of skeletondensity (SD) should be used.

Why only two parameters- vessel density and vessel skeleton density have been used for the analysis? The background is not exhaustingly explained in the introduction section.

The following sentence (line 62-63) is not proper: Inclusion criteria were eyes of diabetic patients without DR, or with mild, moderate or severe NPDR. Eyes cannot be inclusion criteria.

In the methods section it has been written that “Inclusion criteria were eyes of diabetic patients without DR, or with mild, moderate or severe NPDR” but in the abstract and the title it has not been pointed out that only eyes non-proliferative diabetic retinopathy are included. In the abstract it is written that “cohort of diabetic subjects” has been analysed. There is no control group, proliferative diabetic retinopathy is not included at all. Were eyes with macular edema included?

In the figure 4 abbreviations are not explained (For example MM NPDR).

Visual acuity is not included in the analysis.

The conclusions of the study are  not significant for the clinical practice nor for scientific purposes.

Author Response

Answers to reviewers

Dear editor and dear reviewers, thank you very much for the consideration of our manuscript and for your comments.

We answered all your remarks and modified the manuscript as you suggested.

Reviewer 1

The phrase vessel skeleton density (VSD) instead of skeleton density (SD) should be used.

We have replaced the phrase following your recommendations.

Why only two parameters- vessel density and vessel skeleton density have been used for the analysis? The background is not exhaustingly explained in the introduction section.

Our objective was to look for a relation between data easily available on the device usable by all ophthalmologists and the severity of diabetic retinopathy in order to be usable in clinical practice. On most OCTA devices vessel density and vessel skeleton density are easily and automatically measured on the superficial capillary plexus. We have added these specifications in the introduction.

The following sentence (line 62-63) is not proper: Inclusion criteria were eyes of diabetic patients without DR, or with mild, moderate or severe NPDR. Eyes cannot be inclusion criteria.

We have modified the sentence.

In the methods section it has been written that “Inclusion criteria were eyes of diabetic patients without DR, or with mild, moderate or severe NPDR” but in the abstract and the title it has not been pointed out that only eyes non-proliferative diabetic retinopathy are included. In the abstract it is written that “cohort of diabetic subjects” has been analysed. There is no control group, proliferative diabetic retinopathy is not included at all. Were eyes with macular edema included?

We have modified the title and the abstract in order to specify the population.

We did not include patients with proliferative diabetic retinopathy (PDR) because we only have among our patients very few diabetic patients with PDR and without treated DME and PRP or other exclusion criteria.

We did not include a control group, because our idea was not to compare the non-perfusion between diabetic and non-diabetic patients, but to find an association between the non-perfusion and the severity of DR and compare patients with no DR, to patients NPDR. The absence of control group was specified in the limitation chapter of this study.

We have excluded patients with diabetic macular edema under treatment only.

In the figure 4 abbreviations are not explained (For example MM NPDR).

We have added abbreviations to all legends.

Visual acuity is not included in the analysis.

We have added the results of BCVA.

The conclusions of the study are not significant for the clinical practice nor for scientific purposes.

Biomarkers of severity of diabetic retinopathy such as macular or nasal non-perfusion are useful and allow us to assess the severity of the patients without analyzing the entire periphery of the retina. To date, the assessment of the DR on ultra-wide field imaging seems to be more precise than the assessment on 7ETDRS fields, even if studies are still on-going (Protocol AA, EVIRED). All the ophthalmologists will not be able to access UWF imaging because of its cost. However, all ophthalmologists whatever their specialty usually have on OCT and nowadays OCTA are available on most OCT device. Our goal was to look for easy biomarkers associated with the severity of diabetic retinopathy. Moreover, analyzing nasal retina allow us to do measurements even in case of macular edema, or macular exsudates.

Of course, our work is just a preliminary exploration and needs to be confirmed by a multi-center study, including also PDR patients, to look for a relation between data easily available on the device usable by all ophthalmologists and the severity of diabetic retinopathy in order to be usable in clinical practice. On most OCTA devices vessel density and vessel skeleton density are easily and automatically measured on the superficial capillary plexus. We have added these specifications in the introduction.

We hope that the changes we have made following your recommendations will help our manuscript to be more suitable for publication.

Thank you very much.

Best regards,

The authors

Reviewer 2 Report

This is a  prospective study regarding reduced vessel density in the mid-periphery and peripapillary  area of the superficial capillary plexus in diabetic retinopathy.

The topic is interesting and up to date, because the knowledge regarding OCTA parameters in DR is still evolving.

In my opinion the major limitation is the lack of a control group including age and gender matched healthy subjects. Optical Coherence Tomography Angiography (OCTA) is a recently developed, non- invasive, retinal and choroidal imaging technique that provides objective quantification of several retinal and choroidal microvascular parameters in the perifoveal vascular network, such as: vessel density (VD, mm-1), perfusion density (PD, ratio) and foveal avascular zone (FAZ) parameters. Since there is a limited number of reliable data regarding OCTA parameters in normal, healthy eyes, every OCTA study should be presented in comparison to the control group.

Jia, Y.; Tan, O.; Tokayer, J.; Potsaid, B.; Wang, Y.; Jonathan, J.; Kraus, M.F.; Subhash, H.; Fujimoto, J.G.; Hornegger, J.; et al. Angiography With Optical Coherence Tomography. Clin. Sci. 2012, 20, 3116–3121.

Others issues, which also need to be resolved:

  1. Please verify and define specific ocular exclusion criteria. To make sure, that reported OCTA changes are due to DR, other ocular diseases must be excluded. What about ocular surgeries, ocular ischemic syndrome, etc.

  1. General co-morbidities, such as cardiovascular diseases i.e. atherosclerosis or coagulation disorders could have a significant impact on OCT-A results. In my opinion it would be worth considering to exclude those patients with macrovascular complications from the study group.

  1. To be sure, that the methodology of the study is correct, to avoid risk of bilaterality bias, only one eye per patient should be randomly selected. Also in cases with asymmetric stage of DR, usually the eye with higher DR grade should be selected.

  1. Please verify if the results were adjusted for the cigarette smoking status in the study and in the control group.

Latest results suggested, that smoking is likely associated with deleterious changes in the retinal microvasculature of patients with a history of diabetes and no visible DR.

Liu DW, Haq Z, Yang D, Stewart JM. Association between smoking history and optical coherence tomography angiography findings in diabetic patients without diabetic retinopathy. PLoS One. 2021 Jul 9;16(7):e0253928. 

Author Response

Answers to reviewers

Dear editor and dear reviewers, thank you very much for the consideration of our manuscript and for your comments.

We answered all your remarks and modified the manuscript as you suggested.

Reviewer 2

This is a prospective study regarding reduced vessel density in the mid-periphery and peripapillary area of the superficial capillary plexus in diabetic retinopathy.

The topic is interesting and up to date, because the knowledge regarding OCTA parameters in DR is still evolving.

In my opinion the major limitation is the lack of a control group including age and gender matched healthy subjects. Optical Coherence Tomography Angiography (OCTA) is a recently developed, non- invasive, retinal and choroidal imaging technique that provides objective quantification of several retinal and choroidal microvascular parameters in the perifoveal vascular network, such as: vessel density (VD, mm-1), perfusion density (PD, ratio) and foveal avascular zone (FAZ) parameters. Since there is a limited number of reliable data regarding OCTA parameters in normal, healthy eyes, every OCTA study should be presented in comparison to the control group.

Jia, Y.; Tan, O.; Tokayer, J.; Potsaid, B.; Wang, Y.; Jonathan, J.; Kraus, M.F.; Subhash, H.; Fujimoto, J.G.; Hornegger, J.; et al. Angiography With Optical Coherence Tomography. Clin. Sci. 2012, 20, 3116–3121.

Of course, you are right. We did not include a control group, because our idea was not to compare the non-perfusion between diabetic and non-diabetic patients, but to find an association between the non-perfusion and the severity of DR and compare patients with no DR, to patients with NPDR. The absence of control group was specified in the limitation chapter of this study.

Other issues, which also need to be resolved:

  1. Please verify and define specific ocular exclusion criteria. To make sure, that reported OCTA changes are due to DR, other ocular diseases must be excluded. What about ocular surgeries, ocular ischemic syndrome, etc.

 We verified all the medical ocular history of our 56 patients, and none of them had vitreous surgery or ocular ischemic syndrome, or other retinal vascular disease. We have added these exclusion criteria.

  1. General co-morbidities, such as cardiovascular diseases i.e. atherosclerosis or coagulation disorders could have a significant impact on OCT-A results. In my opinion it would be worth considering to exclude those patients with macrovascular complications from the study group.

You are right we should have excluded all the macrovascular complications. However, we have collected all the medical history of included patients, it was self-reported reports. We have retrospectively explored all the data and were able to state that patients suffered from treated high blood pressure, treated sleep apnea, treated dyslipidemia, no other condition was specified. Unfortunatly, we did not access to a common medical record and had to consider the declaration of patients. No patient declared suffering from severe atherosclerosis or coagulation disorder. However, we specified this in the method chapter.

  1. To be sure, that the methodology of the study is correct, to avoid risk of bilaterality bias, only one eye per patient should be randomly selected. Also in cases with asymmetric stage of DR, usually the eye with higher DR grade should be selected.

 We performed a sub analysis considering only one eye per patient as you have suggested. We have added these results in the result chapter.

  1. Please verify if the results were adjusted for the cigarette smoking status in the study and in the control group.

Latest results suggested, that smoking is likely associated with deleterious changes in the retinal microvasculature of patients with a history of diabetes and no visible DR.

Liu DW, Haq Z, Yang D, Stewart JM. Association between smoking history and optical coherence tomography angiography findings in diabetic patients without diabetic retinopathy. PLoS One. 2021 Jul 9;16(7):e0253928. 

We verified that groups were well balanced for smoking status and have added this information in the method chapter.

We hope that the changes we have made following your recommendations will help our manuscript to be more suitable for publication.

Thank you very much.

Best regards,

The authors

Round 2

Reviewer 1 Report

Not all abreviations are explained in the descriptions of figures 7 and 8 (DR, NPDR, ETDRS) , the term "skeleton density" still exists in the descriptions of figures.

Author Response

Answers to reviewers

Dear reviewer, thank you again for your consideration and help in improving this manuscript.

Reviewer 1 :

Not all abreviations are explained in the descriptions of figures 7 and 8 (DR, NPDR, ETDRS) , the term "skeleton density" still exists in the descriptions of figures.

Thank you for noticing. We have added the explanations for abbreviations in figure 7 and 8 and modified also figures 5 and 6 in order to add VSD instead of skeleton density.

We hope that we have answered properly to all the comments of this second round of review.

Best regards,

The authors

Reviewer 2 Report

The authors present results of a prospective study correlating two easily assessed OCTA parameters with  the severity of diabetic retinopathy in order to be usable in clinical practice.

The main significant observation, which was revealed by this study is that, the VD is significantly lower in the nasal peripapillary area and both the VD and VSD are significantly lower in the macular area in patients with more severe DR. This observation may serve as a guide for future OCTA studies to improve their usefulness in everyday clinical practice.

Besides, the topic is interesting and up to date, because the knowledge regarding OCTA parameters in DR is still evolving.

I still think, that the major limitation of the study is the lack of a control group including age and gender matched healthy subjects. However, it was underlined in the limitations of the study and I am willing to accept that.

Others issues, were successfully resolved in the revised version:

  1. Please verify and define specific ocular exclusion criteria.

VERIFYIED

  1. General co-morbidities, such as cardiovascular diseases i.e. atherosclerosis or coagulation disorders could have a significant impact on OCT-A results. In my opinion it would be worth considering to exclude those patients with macrovascular complications from the study group.

VERIFYIED

  1. To be sure, that the methodology of the study is correct, to avoid risk of bilaterality bias, only one eye per patient should be randomly selected. Also in cases with asymmetric stage of DR, usually the eye with higher DR grade should be selected.

DONE, BUT I WOULD LIKE TO ASK TO VISUALIZE THOSE RESULTS USING FIGURES AND PLEASE ATTACH THEM TO SUPPLEMENTARY MATERIALS.

  1. Please verify if the results were adjusted for the cigarette smoking status in the study and in the control group.

DONE

Author Response

Answers to reviewers

Dear reviewers, thank you again for your consideration and help in improving this manuscript.

Reviewer 2:

The authors present results of a prospective study correlating two easily assessed OCTA parameters with  the severity of diabetic retinopathy in order to be usable in clinical practice.

The main significant observation, which was revealed by this study is that, the VD is significantly lower in the nasal peripapillary area and both the VD and VSD are significantly lower in the macular area in patients with more severe DR. This observation may serve as a guide for future OCTA studies to improve their usefulness in everyday clinical practice.

Besides, the topic is interesting and up to date, because the knowledge regarding OCTA parameters in DR is still evolving.

I still think, that the major limitation of the study is the lack of a control group including age and gender matched healthy subjects. However, it was underlined in the limitations of the study and I am willing to accept that.

Others issues, were successfully resolved in the revised version:

  1. Please verify and define specific ocular exclusion criteria.

VERIFYIED

  1. General co-morbidities, such as cardiovascular diseases i.e. atherosclerosis or coagulation disorders could have a significant impact on OCT-A results. In my opinion it would be worth considering to exclude those patients with macrovascular complications from the study group.

VERIFYIED

  1. To be sure, that the methodology of the study is correct, to avoid risk of bilaterality bias, only one eye per patient should be randomly selected. Also in cases with asymmetric stage of DR, usually the eye with higher DR grade should be selected.

DONE, BUT I WOULD LIKE TO ASK TO VISUALIZE THOSE RESULTS USING FIGURES AND PLEASE ATTACH THEM TO SUPPLEMENTARY MATERIALS.

We have added 3 supplementary figures in order to show the results.

  1. Please verify if the results were adjusted for the cigarette smoking status in the study and in the control group.

DONE

We hope that we have answered properly to all the comments of this second round of review.

Best regards,

The authors
